# Strong pathogen competition in neonatal gut colonisation

Tommi Mäklin ●[1] ✉, Harry A. Thorpe ●[2], Anna K. Pöntinen ●[2,3], Rebecca A. Gladstone[2], Yan Shao ●[4], Maiju Pesonen ●[2], Alan McNally ●[5], Pål J. Johnsen ●[6], Ørjan Samuelsen ●[3,6], Trevor D. Lawley ●[4], Antti Honkela ●[1] & Jukka Corander ●[2,4,7] ✉

Opportunistic bacterial pathogen species and their strains that colonise the human gut are generally understood to compete against both each other and the commensal species colonising this ecosystem. Currently we are lacking a population-wide quantification of strain-level colonisation dynamics and the relationship of colonisation potential to prevalence in disease, and how ecological factors might be modulating these. Here, using a combination of latest high-resolution metagenomics and strain-level genomic epidemiology methods we performed a characterisation of the competition and colonisation dynamics for a longitudinal cohort of neonatal gut microbiomes. We found strong inter- and intra-species competition dynamics in the gut colonisation process, but also a number of synergistic relationships among several species belonging to genus *Klebsiella*, which includes the prominent human pathogen *Klebsiella pneumoniae*. No evidence of preferential colonisation by hospital-adapted pathogen lineages in either vaginal or caesarean section birth groups was detected. Our analysis further enabled unbiased assessment of strain-level colonisation potential of extra-intestinal pathogenic *Escherichia coli* (ExPEC) in comparison with their propensity to cause bloodstream infections. Our study highlights the importance of systematic surveillance of bacterial gut pathogens, not only from disease but also from carriage state, to better inform therapies and preventive medicine in the future.

Human gut bacteria are generally considered commensal organisms, but some of them harbour the considerable potential to cause either mild or severe infections outside the gut. One of the most prominent examples is extra-intestinal pathogenic *Escherichia coli*, which is the predominant facultative anaerobe in the large intestine[1]. Work on *E. coli* going back several decades suggests strong intra-species competition in healthy colonisation based on serotypic variation, or the lack

of thereof[2]. Multi-locus enzyme electrophoresis typing studies done on longitudinal collections of stool confirmed these conclusions in the early 1980s[3,4].

Recent "bottom-up" experimental studies further support that inter-species competition plays a key role in shaping bacterial gut communities[5,6]. However, despite considerable research effort over the years on this topic, systematic population-wide characterisation of

[1]Helsinki Institute for Information Technology HIIT, Department of Computer Science, University of Helsinki, Helsinki, Finland. [2]Department of Biostatistics, University of Oslo, Oslo, Norway. [3]Norwegian National Advisory Unit on Detection of Antimicrobial Resistance, Department of Microbiology and Infection Control, University Hospital of North Norway, Tromsø, Norway. [4]Parasites and Microbes, Wellcome Sanger Institute, Hinxton, Cambridgeshire, UK. [5]Institute of Microbiology and Infection, University of Birmingham, Birmingham, UK. [6]Department of Pharmacy, Faculty of Health Sciences, UiT The Arctic University of Norway, Tromsø, Norway. [7]Helsinki Institute for Information Technology HIIT, Department of Mathematics and Statistics, University of Helsinki, Helsinki, Finland. ✉e-mail: tommi.maklin@helsinki.fi; jukka.corander@medisin.uio.no

the colonisation potential and competition dynamics simultaneously across intra- and inter-species levels is still lacking. Our current study aims to address the need to assess these aspects for several of the major human gut species with pathogenic potential.

In animal models the role of intra-species competition in colonisation has been widely studied, producing certain interesting results related to the species that also colonise the human gut microbiome. For example, colonisation by *E. coli* has been shown to reduce the abundance of *Salmonella typhimurium* in the mouse gut through competition for iron[7], and *Klebsiella michiganensis* was recently demonstrated to prevent mouse gut colonisation by *E. coli* in a particular ecological setting[8]. Other examples of intra-species competition include varying colonisation abilities between *E. coli* strains in gnotobiotic mice[9], and subpopulations of *S. typhimurium* within intestinal tissues mediating colonisation resistance against systemic strains through nutrient competition[10].

In the human microbiome, a recent longitudinal study[11] tracked the competition among *E. coli* strains in particular and the microbiome composition in general over multiple years in a single patient suffering from Crohn's disease, consequently having a dysbiotic microbiome and highly dominant abundance of *E. coli*[11]. Several commonly found *E. coli* clones were seen taking the role of the dominant strain over time, but return to a previously identified strain was never observed[11]. In another study[12], the diversity of *E. coli* colonising the gut prior to, during, and after an ecological disruption was highlighted in an enterotoxigenic *E. coli* challenge, but for a small number of test subjects, which makes it hard to draw more general conclusions about the colonisation potential[12]. Characterising the diversity of colonising pathogenic bacteria in the gut is relevant per se, but also interesting in relation to polymicrobial infections, which have not been widely studied to date. In particular, in the urinary tract infection context, it has been shown that multiple pathogen species can form a complex network with both negative and positive interactions[13].

In this study, we address the colonisation potential and competition dynamics questions simultaneously across intra- and inter-species levels. This was done by performing strain identification and genome assembly for a set of species with pathogenic potential using metagenomes collected from a longitudinal cohort of neonatal gut microbiome samples[14]. We chose the particular neonatal cohort for multiple reasons. Firstly, because of its longitudinal aspect, which enabled us to investigate the dynamics of the competition. Secondly, due to its high sequencing depth and state-of-the-art short-read technology used, which enabled us to assemble the genomes of the colonising strains in sufficient detail for genomic epidemiological analysis. Thirdly, the DNA obtained from a sufficiently large number of faecal samples via the culture-independent approach used in the study provided an opportunity for us to screen for the pathogens in an unbiased and robust manner. Finally, the neonatal cohort was ideal for the study since it enabled us to probe the competition dynamics between the pathogen strains and species when they colonise a niche that is approximately empty, unlike the situation later in life. In addition, we screened a large body of microbiome studies to identify other published studies that would be suitable for our purposes, but only the chosen study[14] was deemed appropriate, satisfying all these desiderata.

In the selected study cohort, stool samples were collected from the neonates at 4, 7, and 21 days after birth, later during the infancy period for a subcohort, and also from the mothers of a subcohort during their stay in the maternity unit. The neonates participating in the study represented two delivery cohorts: one for the vaginally born, and the other for caesarean section birth, the latter of which was shown to be associated with a massive shift in the general gut microbiome composition in the original publication[14]. Access to this longitudinal data from metagenomic sequencing at an unprecedented depth (median depth ~20 million paired-end 150 bp reads), combined with recent analytical advances in genomic epidemiology of mixed samples[15,16] and vastly improved availability of large and high-quality genome libraries from studies of key human pathogens based on whole-genome sequencing of single bacterial isolates, enabled us to interrogate the colonisation and competition processes at the level of assembled genomes of pathogen strains.

## Results

### Lineage-level analysis of neonatal gut microbiome data

We analysed 1679 sets of paired-end short reads from gut metagenomes based on stool samples collected from the two delivery cohorts vaginally and caesarean section born babies at several time points (Table 1). Our study extends the previous analysis[14] by providing lineage-level characterization and subsequent genome assembly from these samples for several important pathogen species (Table 2) as well as a more detailed exploration of the diversity within the *Klebsiella* genus. In the remaining text we refer to a "lineage" and a "strain" interchangeably, as is commonplace in the genomic epidemiology literature.

In both the lineage-level characterization and the *Klebsiella* species analysis we applied the recent mSWEEP method for estimating the relative abundances of predefined lineages, or species, from metagenomic short-read sequencing data[15] and the mGEMS method for assigning the metagenomic reads to bins that correspond to a single lineage[16]. The use of both methods was enabled by a bespoke set of reference sequences (described in the "Methods" section) covering the within-species variation in the chosen target species (Table 2). Our intended focus was on pathogenic Enterobacteriaceae and Enterococci, which have been widely considered in colonisation experiments in animal models and have large representative collections of high-quality reference genomes available. However, we did also screen for the presence of multiple other pathogen species but found only sporadic cases of colonisation (Table 2), which prevents drawing any robust conclusions regarding them and they are consequently excluded from further consideration. The analysis pipeline is described in more detail in the "Methods" section and in Supplementary Fig. 1. Comprehensive sets of results from the analyses will be presented in the following sections.

The study cohort in the previously published study[14] consisted of 596 healthy babies from full-term pregnancies, delivered either via vaginal birth (314 babies) or caesarean section (282 babies). All 596 babies provided faecal samples at least once during their first month of life (see Table 1 for details), and 302 babies were resampled once between ages 7–11 months (142 caesarean section and 160 vaginal deliveries). Faecal samples were also available from 175 mothers (65 from the caesarean section cohort and 110 the vaginal delivery cohort), collected before or after delivery in the maternity unit and paired with 178 babies. Since the 175 samples from mothers were not available in a systematic fashion in the original study, we excluded a comparative analysis of the mother vs infant pairs of strains. Further, the isolated genome data (805 samples) from the original study were not included in the analysis of the within-sample variation to avoid biasing our conclusions. This is due to the fact that isolates were cultured in a highly uneven manner, such that a majority of them represented *Enterococcus* (451 out of 805), followed by *Klebsiella* spp. (235/805), while a very small fraction was available for the *Escherichia* spp (41/805). As *E. coli* was found to be the most dominant species in many samples by the metagenomic analysis, the isolate data would not have covered samples systematically. Although more isolates were available for the Enterococci, it was necessary to treat all species in the same manner in our analysis.

### Competition and synergistic relationships drive Enterobacteriaceae and *Enterococcus* colonisation

We first investigated the inter-species competition dynamics between various Enterobacteriaceae and two *Enterococcus* species,

**Table 1 | Number of babies sampled at each timepoint during the first month of life**

| Timepoint | Number of babies sampled | |
| --- | --- | --- |
| (days after birth) | C-section | Vaginal delivery |
| 4 | 153 | 157 |
| 6 | 0 | 2 |
| 7 | 252 | 280 |
| 8 | 1 | 1 |
| 9 | 1 | 1 |
| 10 | 3 | 0 |
| 11 | 0 | 4 |
| 12 | 4 | 4 |
| 13 | 1 | 1 |
| 14 | 2 | 3 |
| 17 | 0 | 1 |
| 18 | 1 | 0 |
| 21 | 178 | 147 |

**Table 2 | List of the target pathogen species, abbreviations of their names used in the manuscript, and the numbers of times lineages of each species were identified reliably in the metagenomics data using the mGEMS pipeline**

| Full name | Abbreviation | Reliably identified colonisations |
| --- | --- | --- |
| *Acinetobacter baumannii* | A. bau | 0 |
| *Escherichia coli* | E. col | 1124 |
| *Enterococcus faecalis* | E. fcs | 690 |
| *Enterococcus faecium* | E. fcm | 38 |
| *Klebsiella aerogenes* | K. aer | 12 |
| *Klebsiella grimontii* | K. gri | 100 |
| *Klebsiella huaxiensis* | K. hua | 1 |
| *Klebsiella michiganensis* | K. mic | 86 |
| *Klebsiella ornithinolytica* | K. orn | 9 |
| *Klebsiella oxytoca* | K. oxy | 27 |
| *Klebsiella pasteurii* | K. pas | 0 |
| *Klebsiella planticola* | K. pla | 4 |
| *Klebsiella pneumoniae* | K. pne | 186 |
| *Klebsiella quasipneumoniae subsp. quasipneumoniae* | K. qpq | 0 |
| *Klebsiella quasipneumoniae subsp. similipneumoniae* | K. qps | 1 |
| *Klebsiella spallanzanii* | K. spa | 0 |
| *Klebsiella variicola* | K. var | 23 |
| *Pseudomonas aeruginosa* | P. aer | 6 |
| *Staphylococcus aureus* | S. aur | 76 |
| *Streptococcus pneumoniae* | S. pne | 0 |

*Enterococcus faecalis* and *Enterococcus faecium*. Using the species-level relative abundance estimates to estimate correlations between the target species, we identified statistically significant ($p < 0.05$, permutation test) antagonistic relationships between *E. coli* and *Klebsiella grimontii, Klebsiella michiganensis*, and *Klebsiella pneumoniae*, and similarly between *E. coli* and *E. faecalis* (Fig. 1). The existence of this relationship is also suggested by the markedly more frequent absence of colonisation by *Klebsiella* species in the vaginally delivered cohort (Fig. 2), where *E. coli* is in contrast more common, and has been previously verified for the *E. coli - K. michiganensis* pair in a mouse gut model[8]. A significant negative correlation was also found between *E. coli* and *Staphylococcus aureus;* however, since the latter species was only present in a limited number of samples and is known to be a common skin coloniser in the groin which could have led to contamination of the samples, no further analysis is conducted on the identified *S. aureus* lineages. The numbers of genomes reliably identified (meaning lineages that can be binned with mGEMS and pass an additional quality check as described in the "Methods" section) at the lineage-level are available in Table 2, providing an indication of the cardinality of our correlation analyses. Of note, large absolute values of a correlation coefficient in metagenomic sequencing of healthy microbiomes would not be expected because the typical (to this setting) low-abundance samples tend to dilute the correlation by signalling a random level association between particular taxa.

Within the *Klebsiella* genus we discovered that several species from the genus had a synergistic relationship with no statistically significant negative correlations observed in either cohort (Fig. 1). Although some of the relationships were retained in both cohorts (*K. grimontii* with *K. michiganensis, Klebsiella oxytoca*, and *Klebsiella pasteurii*), notable differences between the cohorts were observed for the other *Klebsiella* species (Fig. 1). Some of these differences are likely explained by the higher prevalence of *Klebsiella* in the caesarean section delivery cohort (Fig. 2) but for species like *K. pneumoniae* that were commonly found in both cohorts, these observations may be indicative of more complex relationships arising from the different environments.

## Mode of birth affects the gut microbiome composition

Comparing the overall differences in species distribution between the caesarean section and the vaginal delivery cohorts using mSWEEP confirms the results presented in the original study[14]. Namely, *E. coli* is considerably more often found in the vaginal delivery cohort (Fig. 2), while the *Klebsiella* species, *E. faecalis* and *E. faecium* are more common in the C section cohort (Fig. 2). Table 2 shows the overall numbers of times a lineage from each of the species was identified in both cohorts, including screened species that were not identified in any of the samples.

### *E. coli* lineages rarely coexist with *Klebsiella* species or each other

Next, we looked in more detail into the *E. coli* lineage composition and coexistence by analysing co-occurrences of *E. coli* multilocus sequence types (STs) with each other and *Klebsiella* species. We found little overlap, with the majority of the cases containing just one *E. coli* ST or *Klebsiella* species (Fig. 3). When coexistence was observed we did not find it happening in a systematic way, with most identified coexisting pairs or triplets observed just a few times depending on the overall prevalence of the particular types in the data set. Notable exceptions occurred in the case of the *K. michiganensis–K. grimontii* pair and the *K. grimontii–K. pneumoniae* pair, which were found together in the caesarean section delivered cohort a total of six times (out of total 596 samples collected from the C-section delivered babies during the first month of life) each and in the *K. michiganensis–K. grimontii* case were also established as synergistic in the correlation analysis (Fig. 1).

### Neonatal gut colonisation of *E. coli* adheres to the first-come, first-served principle

A more detailed analysis was carried out to scrutinise the possible variation across the time points the *E. coli* lineages appear to colonise the neonatal gut and to what degree they are inherited from the mother. In addition, we examined whether the lineages that are successful within the first 21 days persist in the infancy period sampling 4–12 months later. Examining the colonisation-time trajectories for each individual, we found the colonisation typically to already haven taken place at the very first sampling time at 4 days in the vaginal

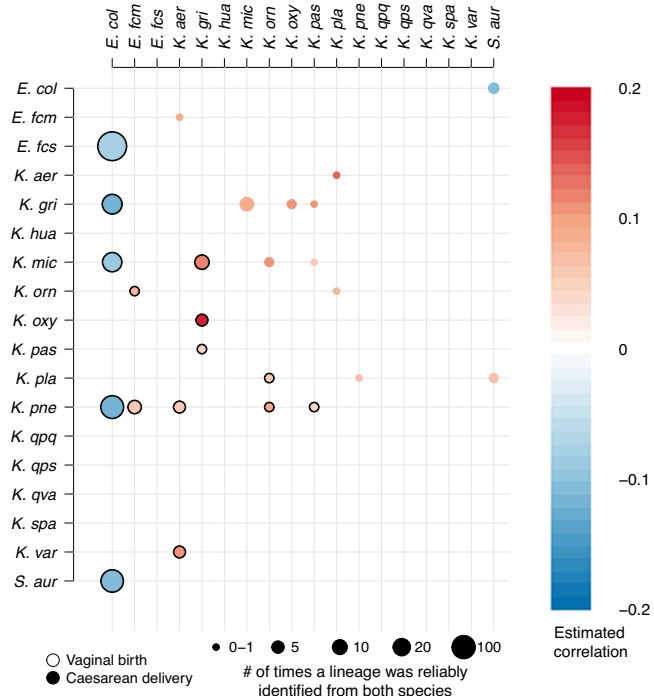

**Fig. 1 | Correlations between the identified priority pathogens.** The figure shows only the statistically significant (*p* < 0.05, two-tailed permutation test) positive and negative correlations for the focus pathogen species. Correlations shown with a black border around the circle are for the vaginal delivery cohort (below diagonal), and correlations without a border for the caesarean section delivery cohort (above diagonal). Darker shades of red and blue represent stronger positive and negative correlation, while areas of the circles are proportional to the number of samples where the correlated pair was reliably identified. The correlation values were estimated from the relative abundance estimates. The exact *p*-values are available in the GitHub repository containing the plotting scripts used.

delivery cohort (Fig. 4a; 88 out of total 314 in the cohort were colonised at 4 days, 184 of whom were detected to carry *E. coli* at some point in the first 21 days). In the subsequent sampling point at 7 days after birth nearly all of the infants who had detectable amounts of *E. coli* at any time point were colonised (150 out of 184; Fig. 4 and Supplementary Fig. 2a) and 66 of the infants were carrying the lineage that was also observed at 4 days. In the remaining 22 individuals out of the total 88 carrying *E. coli* at 4 days, in 3 cases, the lineage identified at 4 days had been replaced by another lineage and in 19 cases, no *E. coli* was present in detectable amounts at 7 days.

Conversely, in the caesarean section delivery cohort there were markedly fewer *E. coli* found overall in the early time points (21 carried *E. coli* at 4 days out of total 282 in the cohort, of whom 97 had detectable amounts of *E. coli* in the first 21 days), with some signs of the initial colonisation happening slightly later at 7 days (59 out of 97; Fig. 4 and Supplementary Fig. 2b). Carriage of the same lineage persisted from 4 to 7 days in 11 infants out of the 21 who carried *E. coli* at 4 days, and in 23 infants from 4 or 7 days to 21 days.

Comparing lineages identified in the mothers to those identified in the infants either at the 4 or the 7 days time point in either cohort revealed 16 infants who shared the same lineage with their mothers (a total of 43 mothers out of the 175 in the full study cohort had detectable amounts of *E. coli*), indicating potential transmission. All of the 16 potential transmissions at 4 or 7 days happened in the vaginal delivery cohort. Transmission in the caesarean section cohort was only observed in the later time points (4 observed transmissions; Fig. 4 and Supplementary Fig. 2b). Based on these results the majority of the initial colonisations in both cohorts appear to have been obtained from an unsampled source rather than transmitted directly from the mother, although we cannot exclude transmission of *E. coli* low-abundance (below the detection limit) lineages in the gut or from other body sites known to sometimes harbour *E. coli*, such as the vagina[17].

Examining the overall course from the first days to the final samplings at 21 days and in the infancy period revealed that, in both

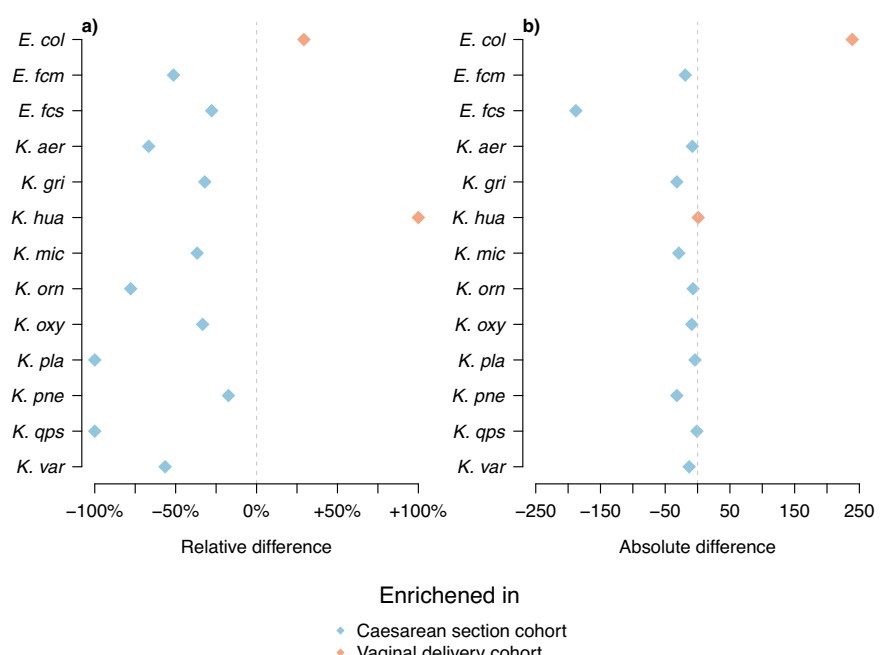

**Fig. 2 | Differences in pathogen loads between cohorts.** The figure shows differences in the number of reliably identified pathogens in each cohort. **a** The relative differences between the cohorts, and **b** the absolute differences.

Pathogens which are more common in the vaginally delivered cohort are coloured in orange and those more common in the caesarean section cohort are coloured in blue.

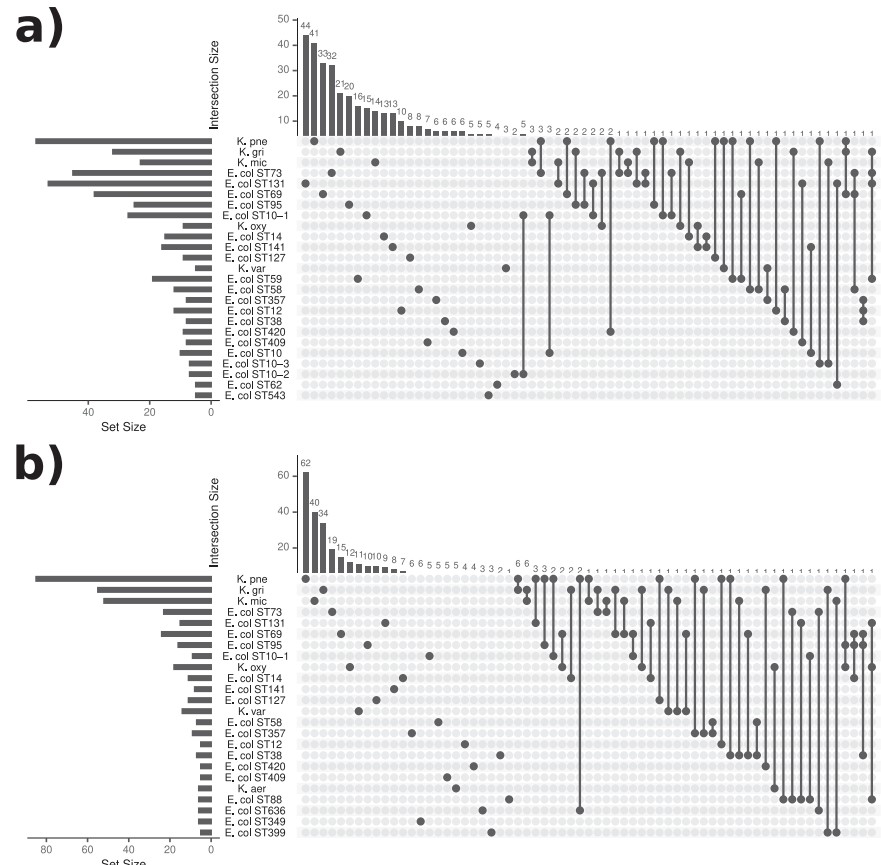

**Fig. 3 | UpSet plot showing coexistence of *E. coli* lineages with *Klebsiella* species.** The plot displays the number of times the *E. coli* lineages and various *Klebsiella* species were found either alone (single dots) or together in a sample (connected dots), and at least five times. Data are shown in (**a**) for the vaginal delivery cohort, and in (**b**) for the caesarean section delivery cohort. Set size (bottom-left subpanels) refers to the number of times a taxonomic unit was found in total, while intersection size (top subpanels) refers to the number of times a taxonomic unit was found alone or coexisting with other unit(s).

cohorts, the *E. coli* lineage that initially colonised the gut at the 4 or 7 days time point either persisted into the final day 21 sampling point or vanished completely (65 colonisations out of total 84 in the 21 day sampling point were already detected at 4 or 7 days in the vaginal delivery cohort; 23 out of 63 in caesarean delivery). Transitions to another lineage within the first 21 days of life were uncommon; only 4 such transitions were observed. When transitions happened, they primarily occurred between the last time point in the first 21 days time series, where *E. coli* was detected, and the infancy period sampling, where a longer period of time had passed (27 such transitions; Supplementary Fig. 2).

Finally, investigation of the samples which were observed to contain multiple *E. coli* lineages (75 in the vaginal delivery cohort and 47 in caesarean, including samples from the mothers) showed that co-colonisation seldom occurred within the first 21 days with only 37 infants being co-colonized at any time point (27 in the vaginal delivery cohort and 10 in the caesarean). In the infancy period the numbers of co-colonised infants increased to 66 (35 vaginal, 31 caesarean). Co-colonisation was also somewhat more frequent in the established microbiomes of the mothers (17 mothers were co-colonised out of the 76 mothers identified carrying *E. coli*). Similar analyses were carried out for the *Klebsiella* species (Supplementary Fig. 3) but they were often detectable only in a single time point despite them being commonly found in the caesarean section cohort (Fig. 2), hindering further efforts to characterise *Klebsiella* persistence. Taken together with the findings from the individual colonisation-time trajectories, these results indicate a substantial competitive advantage for the first strain to colonise the gut which lasts at least through the neonatal period (<1 months of age).

### Transitions from carriage of one lineage to another rarely occur during the first weeks of life

We further detailed the dynamics of transition from carriage of one *E. coli* lineage to carriage of another by constructing an event (transmission or persistence) matrix for the lineages that were observed at least twice across all samples. The event matrix contains the numbers of times a transition from carriage of a specific lineage (rows) to carriage of another lineage or persistence of the same lineage (columns) was observed. The samples from the first 21 days (Fig. 5) showed a strong preference for persistence of the first lineage to colonise the gut, with most of the events occupying the diagonal (persistence of the same lineage between two subsequent time points; 48 cases out of total 51 in the vaginal delivery cohort and 16 out of 17 in the caesarean section cohort). Including the infancy period (Supplementary Fig. 4) results in slightly more variability, especially in the vaginal delivery cohort (Supplementary Fig. 4 panel a), however, the observed events still mostly remain on the diagonal.

### Colonisation potential vs invasiveness of *E. coli* sequence types

We determined the relative invasiveness of *E. coli* lineages by using odds ratios (ORs) for the frequency of each lineage in neonatal gut colonisation compared against the frequency of the same lineages in two systematic disease cohorts of *E. coli* from bloodstream infections: the NORM collection from Norway[18], and the BSAC collection from the UK[19] (see "Methods" for more details). These two cohorts represent the largest, and most importantly, systematic and unbiased genomic studies of *E. coli* bloodstream infections from

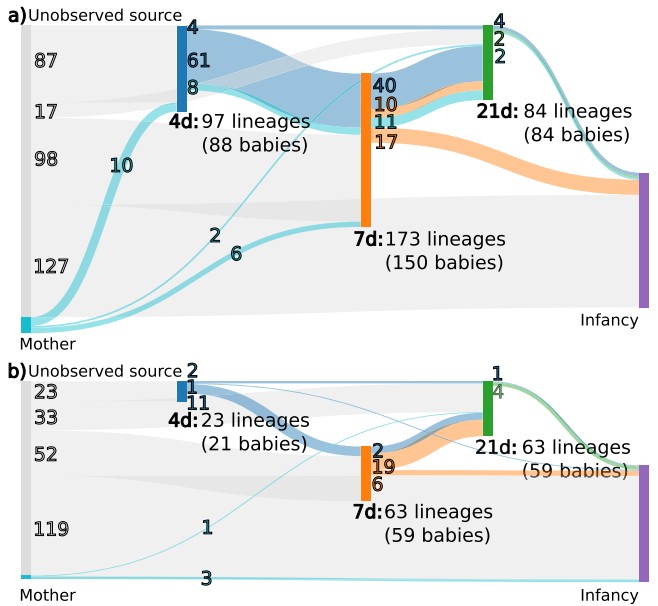

**Fig. 4 | Summary figure of *E. coli* gut colonisation.** The plot displays the numbers of *E. coli* lineages found at each of the five major sampling points (4, 7, and 21 days after birth, in the infancy period, and from the mothers in the maternity unit). The flows indicate the (possible) numbers of lineages that were transmitted between the sampling points, with the colours corresponding to the first time point the transmitted lineages were detected in. Flows that skip a sampling point (such as from 4 days to 21 days) indicate that a lineage returned to detectable levels after the skipped sampling point. **a** Displays the vaginal delivery cohort, and **b** the caesarean section cohort.

numerous hospital districts in each study country (Norway/UK), thus providing the necessary resolution for comparing frequencies in colonisation and disease.

This analysis resulted in confidence intervals for the ORs, which were labelled as either invasive (statistically significant confidence interval with a lower bound higher than 1), commensal (significant with an upper bound <1), or intermediate (confidence interval overlapping 1 or not significantly different from 1). In our results (Fig. 6), the interpretations were not influenced by the choice of the disease cohort, with the exception of the intervals obtained for the multi-drug resistant sequence type ST131 (Fig. 6, Supplementary Fig. 5). The invasiveness of ST131 is overestimated when using the BSAC (UK) data, which was sampled during 2001–2012 when the prevalence of ST131 changed markedly.

We observed that the prevalence in carriage in our neonatal data and carriage in the NORM (Norway) and BSAC (UK) disease collections were similar for ST69 and ST131 but differed for ST73 and ST95, which were significantly overrepresented among the disease isolates from the NORM and BSAC collections (Fig. 6a). In addition to these four nosocomially associated STs, numerous other lineages were observed to have ORs significantly smaller than 1, corresponding predominantly to commensal lineages, for example ST10, with a limited capacity to cause disease.

We also estimated the invasiveness of the major clades of ST131 (A, B, C1 and C2) using the combined disease collections (NORM and BSAC) and found clades A and B to be the most commensal in nature (Fig. 6b). For clades C1 and C2, the ORs were more intermediate with wider confidence intervals. However, when estimated separately using either the NORM (Norway) or the BSAC (UK) collection, the confidence intervals for the C2 clade were particularly affected by which disease collection was used in the comparison. This was likely a result of known differences in prevalence over time and between the UK and Norway[18].

In total, the BSAC (UK) and NORM (Norway) collections contained 83 genomes from clade A, 101 from B, 82 from C1, and 140 from C2.

## Neonatal *E. faecalis* colonisation is not characterised by hospital-associated clusters

We further investigated the specific features of *E. faecalis* colonisation in the delivery cohorts by comparing them to a previously characterised *E. faecalis* species collection[20] using clustering and phylogenetic analyses. While the lineages identified in the neonatal cohort were widely dispersed in the species tree and many of the major STs were well represented among them, the neonates harboured none of the long-term persistent hospital-adapted lineages ST6 and ST9 and contained only a few cases of colonisation by ST28 strains (Fig. 7), which have also been previously defined as hospital-adapted[20]. Furthermore, we found no significant differences ($p > 0.05$, chi-square test) between the delivery cohorts with respect to identified strains harbouring antibiotic resistance-conferring genes to major antibiotic classes (Supplementary Data 1).

Similar to the *E. coli* analysis, we examined the dynamics of transition from carriage of one *E. faecalis* lineage to carriage of another by constructing an event matrix for the lineages that were observed at least twice across the sets of samples (Fig. 8). Again, this analysis revealed a strong preference for the persistence of the first strain to colonise the gut, with a vast majority of the babies not experiencing a switch between sequence types during the first weeks of life and rare appearance of co-colonisation with different sequence types. (Fig. 8, Supplementary Figs. 6, 7).

## Antimicrobial resistance and virulence genes in *Klebsiella* strains

We used Kleborate[21] to detect antimicrobial resistance (AMR) and virulence genes in the 449 *Klebsiella* assemblies obtained from the metagenomic data. We detected no *mcr* or carbapenemase genes and only 5/186 (3%) of the *K. pneumoniae* assemblies harboured a CTX-M-15 extended-spectrum beta-lactamase (ESBL) gene (Supplementary Fig. 8). These assemblies correspond to four individuals (two caesarean and two vaginally born). The assemblies harbouring this gene were assigned to two different sequence clusters (two to sequence cluster (SC) 1, one ST336 and one ST17; and two SC24, both ST323). Both SC1 and SC24 were found in each delivery cohort, and in one individual the same ST336 sequence was observed on both day 7 and 21, indicating sustained carriage. Of the STs detected, at least ST323 has been previously associated with ESBL carriage and nosocomial transmission[22].

Compared to the limited number of AMR genes, we detected a more diverse set of virulence genes that were spread widely across the *Klebsiella* species (Supplementary Fig. 8). There were three assemblies of the hypervirulent *K. pneumoniae* ST23 clone from two individuals (both vaginally born), which harboured colibactin (cbl), aerobactin (iuc), salmochelin (iro), and RmpADC; a suite of virulence factors that are often carried together. Clb2 was carried on ICE*Kp10*, and *iuc1*, *iro1*, *rmp1* were carried together on the virulence plasmid KpVP-1 (confirmed by Kleborate and subsequent mapping). We also observed 32 *K. pneumoniae* assemblies from 12 different SCs that harboured a diverse set of loci of the siderophore yersiniabactin (ybt). We detected other virulence factors that were very common in the more environmental species (ybt in *K. grimontii*, *K. michiganensis*, *K. ornithinolytica*, *K. oxytoca* and iro in *K. aeruginosa*). Although this may seem concerning, we note that these are classified as "unknown" alleles, meaning that although the genes from this locus are present (>90% nucleotide identity and >80% coverage), they are not the same alleles as those found in clinical strains of *K. pneumoniae*. Little is known about the prevalence of ybt and iro loci in these other *Klebsiella* species, but their presence should not be unexpected; a small study previously found that 5/11 isolates in the study (representing 5 different *Klebsiella*

species) carried ybt[23], and another larger study based on whole-genome sequencing data showed that *K. ornithinolytica* commonly carries ybt[24]. More research is needed to explore the evolutionary and clinical significance of these loci.

## Discussion

The colonisation of the gut by opportunistic bacterial pathogens has been a topic of intensive research for decades. Data from observational in vivo studies, human bacterial challenge experiments, and animal models have pointed to both antagonistic and synergistic relationships between various species. However, clear insight into the competition dynamics at both the intra- and inter-species level in the human gut has

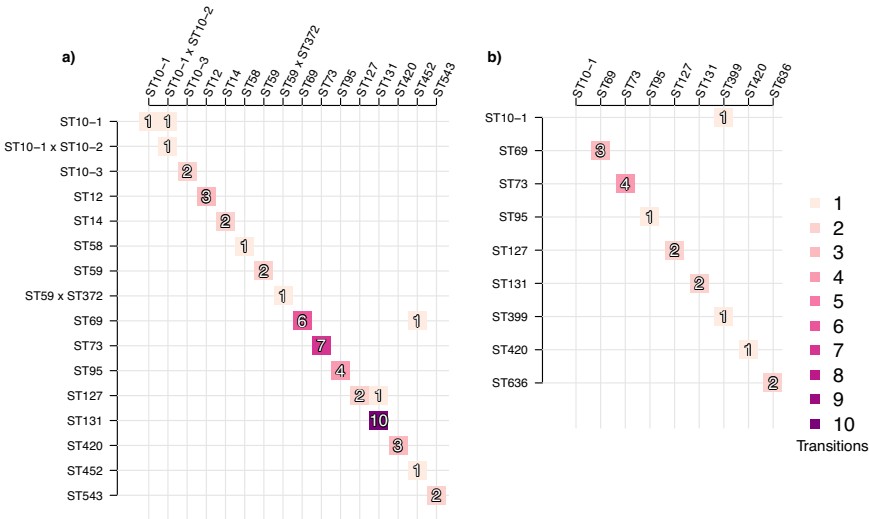

**Fig. 5 | Event matrix displaying colonisation identities with respect to *E. coli* lineages between subsequent time points.** The figure shows events corresponding to either transition from one *E. coli* lineage (rows) to another *E. coli* lineage (columns) or persistence of the same lineage (diagonal). **a** Events for the vaginal delivery cohort with samples from the infancy period excluded, and **b** Events for the caesarean section delivery cohort with infancy period excluded. Darker shades of purple denote more common events, the count of which is also indicated by the number contained within the shaded boxes. Lineages shown were observed at least twice across the whole set of samples.

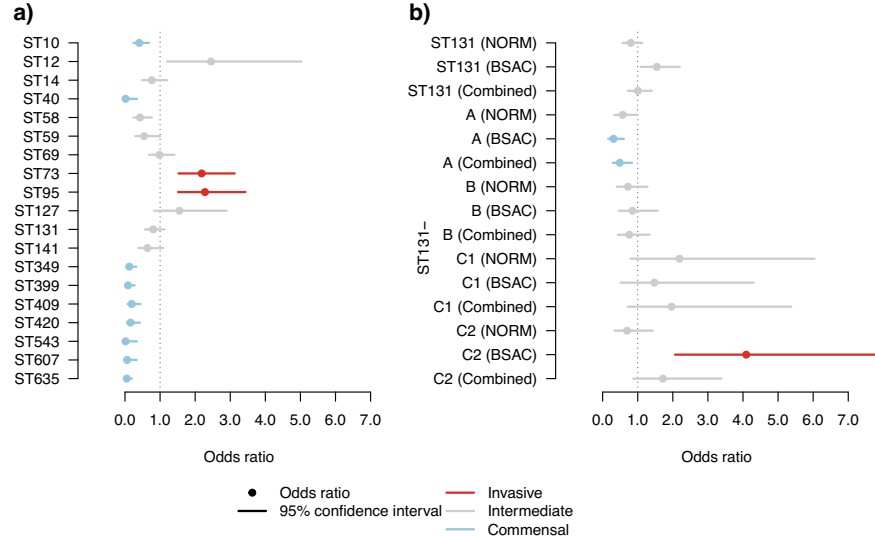

**Fig. 6 | Odds ratios for relative *E. coli* invasiveness.** The odds ratios (OR) for invasiveness are displayed with the 95% confidence interval centred on the OR. The confidence intervals with a statistically significant ($p < 0.05$, two-tailed Pearson's chi-squared test with Benjamini-Hochberg adjustment for multiple testing) lower bounds greater than 1 correspond to more invasive lineages (coloured red; OR statistically significantly higher than 1), and confidence intervals with a statistically significant upper bound <1 to more commensal lineages (coloured light blue; OR statistically significantly <1). Lineages with a confidence interval that contains the value 1, or are not statistically significantly different from 1, are labelled as intermediate (coloured grey). The confidence intervals are shown in (**a**) for both the top 10 most frequent lineages in Norwegian (NORM) bloodstream infections (STs 10, 12, 14, 59, 69, 73, 95, 127, 131, and 141) and all additional lineages, where the OR was estimated to differ from 1 (significantly or not). **b** The confidence intervals estimated for both ST131 as a whole and additionally for its established sublineages (ST131 A, B, C1, and C2). For both ST131 and the subclades, the estimates are reported using either the BSAC (UK) collection, the NORM (Norway) collection, or combined collections for comparison. The exact p-values and the sample sizes are available in the Source Data file for this figure.

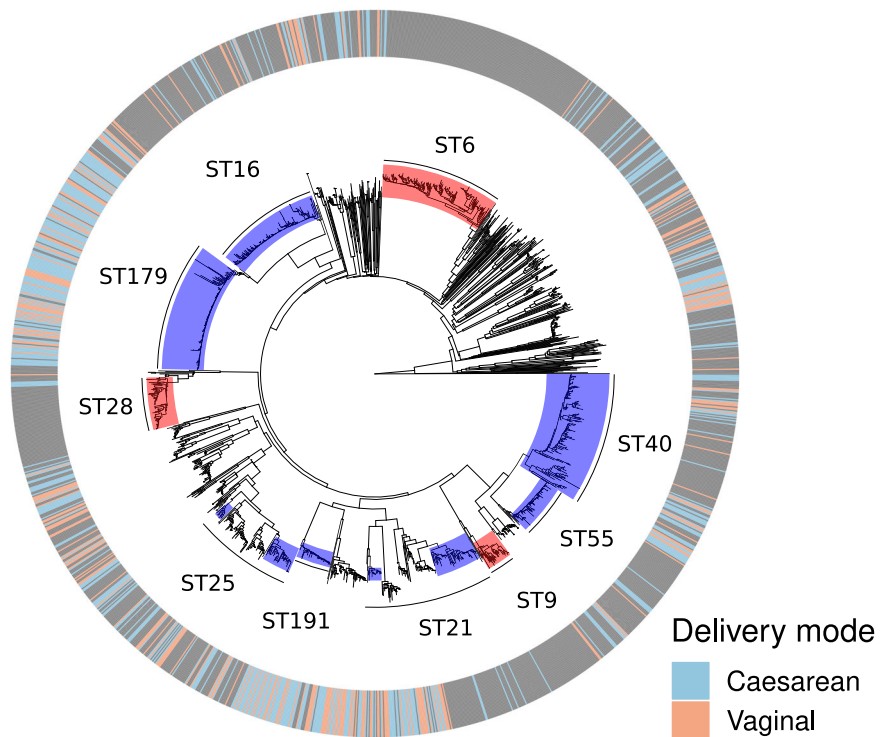

**Fig. 7 | Embedding of the neonatal cohorts within a general *E. faecalis* species-wide collection.** Outer metadata blocks depict the delivery mode of the cohorts (caesarean, light blue; vaginal, orange), aligned against the neighbour-joining (NJ) phylogeny from the core distances. Ten largest sequence types in the combined collections are highlighted within the branches, and the sequence types previously defined as hospital-adapted[20] are coloured in red.

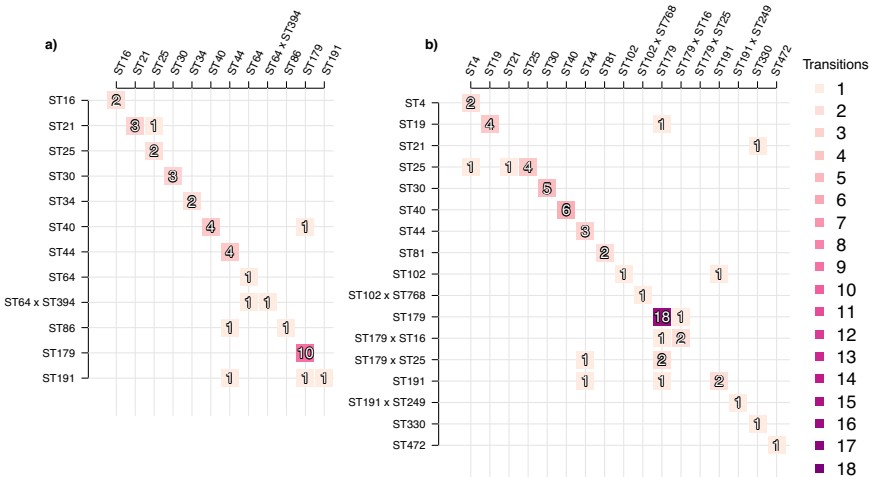

**Fig. 8 | Event matrix displaying colonisation identities with respect to *E. faecalis* lineages between subsequent time points.** The figure shows events corresponding to either transition from one *E. faecalis* lineage (rows) to another *E. faecalis* lineage (columns) or persistence of the same lineage (diagonal). **a** Events for the vaginal delivery cohort with samples from the infancy period excluded, and **b** Events for the caesarean section delivery cohort with infancy period excluded. Darker shades of purple denote more common events. Lineages shown were visited at least twice across the whole set of samples.

been missing in particular for the early phase of life when the gut microbiome opens up for colonisation. Here, we were able to advance this understanding thanks to the deep sequencing of neonatal stool samples in a previous landmark study[14]. Combined with recent methodology[15,16] and high-precision genomic reference libraries, these results allowed us to identify and assemble single genomes from metagenomic sequencing data at the level of resolution required for standard bacterial genomic epidemiology, including phylogenetic analysis. The degree of resolution constitutes the main difference between our analysis and other available standard reference genome-based metagenomics approaches, which either do not allow extracting the assembled genomes[25] or do not perform well when strain-level variation is present in the samples[26,27].

As the purpose of our study was to unravel the competition dynamics of gut colonisation for selected key human pathogens, any possible implications for the actual host health, either during infancy or later in life, are left out of the scope of this study. For the same reason, we do not reflect on our findings in light of the extensive literature on human microbiome variation in a more general context. Nevertheless, it is important to note that none of our focal pathogens

are expected to cause disease in young children, unless they were immunocompromised or underwent surgery which would significantly increase the risk of the bacteria entering the bloodstream.

We found no or very few examples of nosocomially adapted lineages of *K. pneumoniae* and *E. faecalis* in the neonatal samples. Moreover, the few cases we did detect were found evenly distributed between the two delivery cohorts, suggesting that neither group is preferentially colonised with such organisms despite differences in the length of hospital stay and opportunities to become colonised during birth. In particular, we saw no difference between the two cohorts in terms of the frequency of virulence or antibiotic resistance elements called from the assembled genomes of the identified *K. pneumoniae* and *E. faecalis* strains. This is despite the massive microbiome shift observed in the original study[14] that resulted in a stunted microbiome of the caesarean-born babies. Taken together, this suggests that in the absence of a strong selective pressure stemming from the use of antibiotics, the hospital-adapted lineages are generally at a disadvantage when attempting to colonise healthy individuals in the hospital environment. Alternatively, the hospital-adapted lineages may have adapted to succeed in the presence of a mature microbiome and a corresponding metabolome, implying that they are strong competitors but poor pioneers. Overall, the colonisation process of a sterile infant gut appears stochastic without strong selection for the pioneering lineages, and the lineage pool associated with birth in the hospital environment differs from the known nosocomial pool that is typically sampled under antibiotic pressure.

For the most abundant organisms, we were able to identify a large total number of different lineages present across the birth cohorts. Despite all this diversity, only rarely were two or more lineages of the same species detected in any single individual at the same time point, and even transition from carrying one lineage to another lineage in the next time point was uncommon. This implies that the lineages generally compete strongly for the colonisation opportunity and the process in neonates adheres to the 'first come, first served' principle. This stands in stark contrast with the ESBL *E. coli* colonisation dynamics study performed among healthy volunteers at a hospital in Laos, where different clones were frequently replacing each other[28].

Surprisingly to us, we detected abundant diversity of species from the genus *Klebsiella*, which was particularly marked in the caesarean section delivery cohort. These differences in diversity between the cohorts were also reflected in the antagonistic relationship between *E. coli* and several of the *Klebsiella* species, the former of which was much more dominant in the vaginal delivery cohort. In both cohorts, *K. pneumoniae* was the most abundant colonising *Klebsiella* species, but it is typically considered a low-abundance member of the healthy gut microbiome and with a relatively small population prevalence[29–31] dependent on geographical location[32]. A recent large cross-sectional study also found 16% carriage rate of *K. pneumoniae* in a general adult cohort based on selective culturing approach[33]. In our study, the high frequency and diversity of *Klebsiella* detected in the neonates implies that these species are likely to be commonly present in the adult gut to allow for such an ample transmission to the babies, but may remain undetectable with the gold standard genus-specific culturing based approach. Furthermore, similar additional variation might be hidden for *E. coli*, such that additional lineages could be present in the gut but their growth is hindered by the dominating strain(s), and thus they remain under the detection limit with the currently used sequencing depth. We do acknowledge the possibility of transmission of *Klebsiella* and *E. coli* from other body sites, however, for both species, the gut is the primary site of colonisation and thus, colonisation at other sites likely plays a minor role in transmission.

Our present study opened the possibility to systematically compare the colonisation abilities of *E. coli* lineages with their ability to cause bloodstream infections. Despite that such infections mostly happen in the elderly, the neonatal colonisation and competition processes are assumed to be reflective of the population colonisation frequencies of the *E. coli* lineages in the adult gut, since diverse environmental exposure (rather than vertical transmission from mothers) is expected to be a major source of strain transmission during early life and has been observed in several pathobiont species colonising the infant gut[34,35]. Previous attempts in the literature to estimate the level of relative invasiveness (as measured by the odds ratio of a bloodstream infection vs colonisation event) of particular strains in this respect appear to be scarce. Here we used for comparison both a Norwegian[18] and a UK[19] genomic cohort of *E. coli* bloodstream infections that were collected in a systematic manner and representative for the frequencies of lineages in the disease population.

Comparisons of the relative frequencies in colonisation vs disease revealed that the relative invasiveness of the multidrug-resistant *E. coli* ST131 was not particularly high, but its frequency in disease was overall more reflective of that in colonisation. This demonstrates the importance of matching location and time when calculating ORs for invasiveness when prevalence is known to vary temporally or geographically. Interestingly, when comparing the odds ratios at the level of sub-lineages of ST131, we found that their observed relative invasiveness was consistent with indirect estimates inferred from the phylogenetic expansion modelling in a previous study[18], with sublineages A and B determined as less invasive than sublineages C1 and C2. Out of the other successful *E. coli* sequence types frequently found in bloodstream infections, only ST73 and ST95 were found to have a significantly elevated level of invasiveness when contrasting with their estimated frequencies from our neonatal colonisation data. In addition, ST12 showed an elevated OR but this was not statistically significant when corrected for multiple testing.

Another recent study investigated the polymicrobial nature of recurrent urinary tract infections (rUTIs) and found that *E. coli* populations in the gut and the bladder were comparable between women with and without a history of rUTI in both relative abundance and phylogroup[36]. However, a deeper lineage level analysis, such as the one performed here, combined with screening of genetic determinants of persistent bladder colonisation from the metagenome assemblies could provide a more refined characterisation of the possible population differences between gut and the bladder. In addition, since the bladder is a more resource-poor niche compared with the gut, it is possible that the strain-level competition in the colonisation process as observed in our study would be even stronger in the bladder environment, where both efficient iron acquisition systems[37] and oxygen stress tolerance will be of heightened importance[38]. Such an analysis would nevertheless require substantially increased sequencing depths to enable high-resolution genomic epidemiology.

Added benefits of deep sequencing in analysis of heterogeneous colonisation have also been demonstrated in study of within-host diversity in pneumococcal carriage[39]. Using a similar approach to ours on sequenced DNA from plate sweeps, deep sequencing enabled investigation of implications in disease, analysis of the evolutionary responses to antibiotic treatment, identification of lineages present in samples, and performing resistance and other variant calling[39]. Given the expected continuing decrease in sequencing cost and the increased availability of relevant genomic reference libraries, we anticipate that genomic epidemiology from mixed samples, either based on direct metagenomics or on enriched or targeted DNA will offer wide future opportunities to improve our understanding of pathogen persistence, transmission and evolution.

## Methods

### Ethical statement

The study used sequencing data from a previous study[14]. Collecting the data in the original study was approved by the NHS London - City and East Research Ethics Committee (REC reference 12/LO1492) and

written informed consent was obtained from the mothers for their participation and the participation of their children. No separate approval was required for the analysis of the data in our study.

## Sequencing data

We used sequencing data from a previous study[14] that has been published in the European Nucleotide Archive under accession numbers ERP115334 (whole-genome shotgun metagenomics sequencing data) and ERP024601 (isolate sequencing data). The isolate sequencing data was only used as reference data for the reference-based methods. The WGS metagenomics data was the basis for all reported results.

## Reference sequences

We combined 805 isolate assemblies from the source study for the sequencing data[14] with a bespoke reference database containing sequences for priority pathogens (Table 2) and common commensal and contaminant species. For the species that were analysed in more detail (*E. coli, E. faecalis, Klebsiella* species), we constructed additional species-specific assembly collections, which were used to explore the within-species diversity. The *E. coli* species database consisted of data from two studies: a curated collection of ~10,000 *E. coli* sequence assemblies[40] augmented by additional ~3300 assemblies from a Norwegian bloodstream infection collection[18]. For *E. faecalis*, we used ~2000 assemblies from several European countries[20], and for the *Klebsiella* species ~3000 assemblies collected in Italy[24]. The isolate assemblies from the source study[14] belonging to *E. coli*, *E. faecalis*, or *Klebsiella* species were also included in the respective collections.

We also ran the bacterial species identification method MetaPhlAn (v3.0[41], default options) on the metagenomics reads to check for species that had no representation in our already collected reference sequences. For these species, we downloaded the reference and representative genomes available in the NCBI database as of 30 October 2021 and added them to our reference. After collecting the full reference database as described, all of the sequences were processed with a script (available from https://github.com/tmaklin/baby-microbiome-paper-plots [42],) that concatenated sequences consisting of several contigs by adding a 300 bp gap (twice the read length in the reads) between the contigs, and collected the concatenated sequences in a single multifasta file.

## Pseudoalignment index construction

We first indexed the complete multifasta file with the Themisto short-read pseudoaligner (v2.1.0[16], no-colours option enabled) and used Themisto again (load-dbg and colour-file options enabled) to colour the resulting index according to the species designation of the reference sequences. Colouring the index in this manner means that a pseudoalignment to possibly several reference sequences of the same species is reported simply as a single pseudoalignment to somewhere within the species.

For the species in the priority pathogens list (Table 2), we extracted the reference sequences belonging to each of the priority pathogen species from the multifasta file, and used Themisto (default options) to build additional species-specific pseudoalignment indexes for these target species. In this index, we did not incorporate any colours, instead representing each sequence individually, which is the format required for the further within-species analyses.

## Reference sequence grouping

For each species-specific set of reference sequences, we next used the PopPUNK[43] clustering method for bacterial genomes to identify lineages within the species that (roughly) correspond to clonal complexes. This was done by running PopPUNK (v2.4.0) multiple times for each species-specific set of reference sequences with several different options (bgmm model with the number of components K ranging from 2 to 32, the dbscan model, and model refinement for all produced bgmm models and the dbscan model), and selecting the model that had both high quality metrics as reported by PopPUNK and was also deemed consistent with the MLST designations for the species in a manual inspection. The options that produced the chosen clustering for each species are included in Supplementary Table 1.

## Lineage identification

Next, we began processing the WGS metagenomics dataset with a hierarchical approach consisting of an initial species detection step and a subsequent lineage analysis step performed for the species in the priority pathogens list. In the initial species detection step, the metagenomics reads were first pseudoaligned[44] with Themisto (v2.1.0, reverse complement handling and output sorting options enabled) against the previously constructed coloured pseudoalignment index, which contained reference sequences for all species considered in our analysis (described in the preceding Reference sequences section). These pseudoalignments were used as input to mSWEEP, which is a method for estimating the relative abundances of some predefined groups in short-read sequencing data[15]. The output from mSWEEP (we used v1.6.0[45], with the write-probs option enabled) was further processed with mGEMS (v1.2.0[46], default options), which is a method for partitioning a set of metagenomics reads into several sets of reads (called bins), each corresponding to one of the groups that were used as the basis for relative abundance estimation in mSWEEP[16]. In this initial species detection step, the groups used were the species and we used mGEMS to extract the bins for all species in our target pathogen list.

In the subsequent lineage analysis step, we analysed the species-specific read bins created in the species detection step. If a read bin had been created for one of the species belonging to the priority pathogens list (Table 2), we pseudoaligned the reads in the bin with Themisto (v2.1.0, reverse complement handling and output sorting options enabled) against the corresponding species-specific pseudoalignment index (constructed as described in the Pseudoalignment index construction subsection). These pseudoalignments were then processed with mSWEEP, this time using the lineage definitions from PopPUNK as the target groups, to estimate the relative abundances of the lineages in the reads assigned to the species bin. We then ran mGEMS on the results from mSWEEP, creating another set of read bins that now corresponded to the lineages identified at a relative abundance greater than 0.01. Finally, we ran the quality control tool demix_check (commit 18470d3[47],) on the resulting lineage-level bins. This tool uses the mash[48] distances between the reads assigned to each bin to provide confidence scores for whether the reads in the bin originate from the lineage they're assigned to, or if the bin was spuriously created due to the actual lineage missing from the reference data. The results from demix_check were used to filter out lineage-level bins that received a low or lowest confidence value (3 or 4 in the demix_check output). The bins that received a high or very high score (1 or 2) are called "reliable identifications" in this manuscript, and the list of these is provided in Supplementary Data 2.

## Assembly

We assembled both the isolate sequencing data for their inclusion in the reference sequence collection, and the lineage-level bins obtained from the WGS metagenomics reads using mSWEEP and mGEMS as described above. The same assembly approach was used for both the isolate data and the lineage-level bins. Before assembly, the reads in question were quality controlled and corrected with the fastp preprocessing tool for fastq files (v0.23.1[49], default settings). The corrected reads were then assembled with the shovill assembly pipeline (v1.1.0[50], with read correction disabled).

## Correlation estimation

We used the FastSpar correlation estimation tool for compositional data (v1.0.0[51,52], 1000 iterations with 200 exclusion iterations) to infer the correlations between species-level abundances for the WGS metagenomics data obtained with mSWEEP as described earlier. Since FastSpar requires operational taxonomic units (counts) as the input data, we converted the output from mSWEEP to count data by multiplying the relative abundance estimates for each WGS metagenomics sample with the total number of reads reported as pseudoaligned by Themisto. Statistical significance of the resulting correlation values was calculated using the permutation test functionality from FastSpar (10 000 permutations with 5 FastSpar iterations per permutation) and significance was determined at the $p < 0.05$ level.

## Odds ratios for invasiveness

As the colonisation collection[14] sampled mother-infant pairs multiple times, the collection does not represent independent sampling. Because of this, we pooled the presence of a lineage in a mother-infant pair for all time points for the invasiveness analyses. Pop-PUNK clusters were assigned (as described earlier) to the combined NORM collection (Norwegian bloodstream infections from 2002 to 2017[18],), the BSAC collection (UK bloodstream infections from 2001 to 2012[19]), and the colonisation collections presented in this paper's source study (healthy newborns sampled during 2014–2017[14],), allowing the relative frequencies of lineages to be compared between carriage and disease. We only compared lineages which were identified more than one time.

In estimating the odds ratios and their confidence intervals, for tables with any zero values, 0.5 was added to all cells in the table before calculating the odds ratios according to established methodology[53]. Statistical significance of the confidence intervals was determined with Fisher's exact test for tables with cell value <5, and otherwise the chi-squared test was used. An adjustment for multiple testing was made using the Benjamini-Hochberg method[54]. In the ST131 sublineage analysis, SKA[55] was used to generate an alignment and a subsequent tree embedding the ST131 carriage assemblies from this study and the ST131 assemblies from both the NORM (Norway) collection[18] and the BSAC (UK) collection[19], from which clade membership of the assemblies in this study could be inferred.

## Comparative analysis of *E. faecalis* population structures

We used the query option from PopPUNK (v.2.2.0[43],) to embed the *E. faecalis* assemblies from the neonatal cohort into an alignment-free clustering of *E. faecalis* sequences from another collection[20]. This allowed us to compare the differences between these two collections by unifying the clustering across them. Then, we constructed a comparative neighbour-joining phylogeny of both the neonatal cohorts and the other *E. faecalis* collection[20] from core distances estimated with PopPUNK[43]. Multi-locus sequence types were retrieved from the assemblies using fastMLST v.0.0.15[56], and antibiotic resistance profiles screened using AMRFinderPlus v.3.10.18[57] against the NCBI database (version 2021-12-21.1) with the '–plus' option enabled and with minimum identity of 75% and minimum coverage of 80%. The differences in the presence of genes conferring resistance to major antibiotic classes (aminoglycosides, macrolides, lincosamides, tetracyclines, and phenicols) between the different delivery modes (caesarean vs vaginal) were compared by using Pearson's chi-squared test with Benjamini-Hochberg adjustment for multiple testing in R v.4.2.0[58]. Vancomycin (glycopeptide) was omitted as none of the assemblies from the isolate sequencing data in the neonate source study[14] harboured *van* genes.

## Detection of AMR and virulence genes in *Klebsiella* strains

We used the Kleborate tool (v2.1.0[21]) to detect and screen for AMR and virulence genes in the assemblies created from the mGEMS lineage-level bins belonging to the *Klebsiella* genus. Compared to other approaches such as AMRFinderPlus, Kleborate is a specialised tool for genomic profiling of sequences specifically from the *Klebsiella* genus that leverages a curated database for this particular genus.

## Statistics and reproducibility

We used all sequencing data provided in the previous study[14]. No data were excluded from the analyses. No statistical methods were used to predetermine the sample size. The experiments were not randomized and the Investigators were not blinded to allocation during experiments and outcome assessment.

## Reporting summary

Further information on research design is available in the Nature Portfolio Reporting Summary linked to this article.

## Data availability

The species-level pseudoalignment index for Themisto v2.1.0, the *E. coli*, *E. faecalis*, and *Klebsiella* species-specific indexes are all available at Zenodo (species-level index https://doi.org/10.5281/zenodo.6656881, *E. coli* index https://doi.org/10.5281/zenodo.6656897, *E. faecalis* index: https://doi.org/10.5281/zenodo.6656903, *Klebsiella* index: https://doi.org/10.5281/zenodo.6656911). Results from the mGEMS pipeline which were assigned high or very high confidence scores using demix_check, forming the core of the analyses presented, are listed in Supplementary Data 2. For all figures presented in the manuscript, source data are provided as a Source Data file. The sequencing data used in this study are available in the European Nucleotide Archive under accession codes ERP115334 (whole-genome shotgun metagenomics sequencing data) and ERP024601 (isolate sequencing data). Source data are provided with this paper.

## Code availability

Figures 1–3 and 5–8 were created using R v4.0.5[58]. Scripts used to create the visualisations and batch job scripts containing the commands and options used to run the analyses are available from GitHub at https://github.com/tmaklin/baby-microbiome-paper-plots [42]. The UpSet plot[59] in Fig. 3 was created using the UpSetR package (v1.4.0[60]). Figure 4 was created using SankeyMATIC (https://github.com/nowthis/sankeymatic).

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

## Acknowledgements

The authors wish to thank the Finnish Grid and Cloud Infrastructure (FGCI) for supporting this project with computational and data storage resources. J.C. and H.A.T. were funded by ERC grant no. 742158 and J.C. additionally by NFR grant no. 299941 and Academy of Finland EuroHPC grant. J.C. and A.H. were supported by the Academy of Finland Flagship Finnish Center for Artificial Intelligence FCAI. R.A.G. and A.K.P. were funded by the AMR grant from Trond Mohn Foundation. Y.S. and T.D.L. are supported by the Wellcome Trust (206194 and 108413/A/15/D).

## Author contributions

T.M., A.H., and J.C. conceived the study design and planned the analyses. T.M., H.A.T, A.K.P., and R.A.G. gathered the reference sequences, clustered them, and constructed the indexes required for the lineage identification analyses. T.M. performed the lineage identification analyses, sequence assembly, correlation estimation, and gathered the results for further analysis. R.A.G. estimated the odds ratios for *E. coli* invasiveness. A.K.P. and M.P. performed a comparative analysis of *E. faecalis* population structure. H.A.T. performed the AMR and virulence gene detection in *Klebsiella* strains. T.M., H.A.T, A.K.P., R.A.G, Y.S., M.P., A.M., P.J.J., Ø.S., T.D.L., A.H., and J.C. participated in interpreting the results and writing the manuscript. All authors read and approved the manuscript.

## Competing interests

The authors declare no competing interests.
