## [Peer review file · Nature Communications]

REVIEWER COMMENTS

Reviewer #1 (Remarks to the Author):

Major comments:

1. While the manuscript is entitled “strong pathogen competition in neonatal gut colonisation” the paper reads more as a “proof of concept” for the proposed methodology (combining metagenomics with strain-level genomic epidemiology). I would propose either to change the title to reflect the proposed methodology or expand and dedicate a paragraph on the neonatal microbiome in the introduction. In any case it should be clarified why the selected cohort (why the neonatal microbiome) was chosen.
2. Directly related to the previous comment. Several conclusions are made in the study regarding the neonatal microbiome, including comparison between birth-mode. However, no discussion or comments are made regarding the importance of these findings or its implication. Just one example, several *Klebsiella* species are found dominant in the vaginal delivery cohort, however what implications does this have on the development of the neonatal microbiome, or health later in life.
3. While the manuscript mentions the “population-wide quantification of strain-level colonisation dynamics for many common bacterial pathogens...”, the study seems to primarily focus on *Enterococcus* and *Enterobacteriaceae*, specifically colonisation of different *Klebsiella* species and *E. coli*. Was it decided beforehand to focus on these species, or was a comprehensive analysis done before hand and these were the only species of interest found within the neonatal microbiome? In any of the two scenarios please describe so within the manuscript.
4. The manuscript lacks any clear description of the used cohort. Although in different points in the manuscript different numbers of samples and timepoints are given this only provides more confusion since it is unclear how many samples/timepoints are present, nor how many samples in which timepoint. It may be a good idea to add a clear description at the start of the result section “Lineage-level analysis of neonatal gut microbiome data” and include the nr. of samples of each cohort/group at each individual timepoint. Maybe include a summary figure.
5. An antagonistic relationship between *E.coli* and different *Klebsiella* species has been reported in the results (L127-131). Although significant ($p < 0.05$), the correlation between said pathogens ranged to max -0.2. I’m not very well versed in cut-offs for correlations, but would you still consider this antagonistic?

6. While the methods section is overall extensive, please provide additional, and if possible, code, of the specific settings used in the analyses. For instance, the script mentioned in L460. In L478-482, different options were used and the result with both high-quality metrics and was restively consistent with the species; MLST designations. Please, provide additional information and note which option was finally chosen. Additionally, different tools such as mSWEEP and mGEMS was used, for the reader who may not be familiar with these tools it may be interesting to include a description of these tools.

7. The used approach is described as a “combination of latest high-resolution metagenomics and strain-level genomic epidemiology methods leveraging large genomic reference libraries”. To fully appreciate this methodology could you please include a description of said method would include and how this would differ from any other reference sequence based approach?

Minor comments:

8. L92. mentions “de-mixing metagenomes”, please provide a description on what could be understood of this.

9. L99. “unprecedented depth”, please mention what depth this would be.

10. For figure 2, I would advise to create two different panels representing each in turn the absolute and relative differences. As it is, the figure is a bit confusing.

11. L166. 6 times out of how many? Please provide some additional information.

12. L169-L189. While interesting, this section is a bit confusing to read. For a large part this is since the cohort has not been properly introduced before (comment 4). For readability, it may further be good to include a concise notation of the different timepoints and the relevant results at each time point. Additionally, for clarity it may be good to use percentages instead of numbers. i.e. 28% instead of 88/314.

13. L190-L194. Again, to improve clarity, it would be better to first report on the mothers (43 out of how many?) and then note that 16 children of these mothers were found to share the same lineage.

14. L202-203. Please provide numbers.

15. L206-L207. Which timepoints would this be?

16. L209. You may have given this earlier, but please provide again the number of samples which were observed to contain multiple E.coli lineages.

17. L200-L220. Although supplementary figures are provided for this section, the results may be cleared and better to follow if a (summary) figure was included within the main manuscript.

18. L227-L228. Please provide a paragraph explaining the event matrix.

19. L237. Please provide a short description of the odds ratio and how this was calculated.

20. L236-L257. Please provide additional description of why said cohorts were chosen. For L247. Would that be the prevalence in carriage within the neonatal dataset?

21. Figure 8 is quite difficult to decipher. Please consider replacing the plot with a table, or else increase the size.

22. To identify AMR within the klebsiella, have you used any other tools than Kleborate? I'm not familiar with Kleborate, but I know for instance that rgi from the card-database has demonstrated better performances than AMRFinderPlus from NCBI.

23. L298-L312. Speaks about "isolates", while in L289 "assemblies" are mentioned. In this section (L287 and down), where specific isolates used or reconstructed genomes from the metagenomic samples?

24. L330. Please define "novel methodology"

25. L342. "common pathogenic gut bacteria", in this case would be Klebsiella species in specific?

26. L391-396. Please provide additional information on the cohorts. How many samples, etc. Please also provide a description of what you would consider “relative invasiveness” and how you would determine this.

Reviewer #2 (Remarks to the Author):

Mäklin and colleagues present an investigation of the prevalence, colonization potential, and competition dynamics of pathogenic gut species in a previously published cohort of gut metagenomes and isolates obtained from infants born by vaginal or C-section delivery. They highlight some co-presence and co-exclusion patterns of noteworthy pathogens, expose the dynamics of *E. coli* strains colonization in the newborns' gut (including the rare strain substitution), and evaluate the invasiveness of selected STs in light of larger cohorts. Additionally, they rule out that *E. faecalis* strains colonizing the newborns' gut are the long-term persistent hospital-adapted ones and survey the antimicrobial and virulence repertoire of their *Klebsiella* strains.

Despite the great amount of data to analyse from the previous publication, this work fails to address key questions in the field like exposing the maternal route for the colonization events (there are mothers' samples that are also presented or even highlighted here and there but never properly investigated and discussed) and presents a very shallow biological understanding and discussion of the results of the performed computational analyses. Moreover, the results presented here are not discussed in light of the enormous amount of literature in the field, and the quality/curation of the figures is low. Methods are not clear enough to ensure reproducibility, and there are many convoluted parts. Nevertheless, the content of this work might be of interest to a more specialized audience.

Here are the major issues that should be addressed:

1) The authors state that they are surveying to what degree *E. coli* is inherited from the mother (Line 173) but they only report lineage sharing data. Such analysis should be done by comparing strains carried by the mother with those carried by the newborn, and to me, it is not clear why the authors did not attempt this (especially because they have isolates as well). On this line, the authors use the word "strain" multiple times in the text to refer to lineages, I guess? Otherwise, there is no explanation of how they performed the strain-level analyses in the Methods. I understand that this might only be a matter of the use of the language, but please clarify.

2) In general, this manuscript is of interest mainly to clinical microbiologists, who might not be familiar with the proposed methodologies developed by the authors in previous publications. I would suggest expanding the Methods but also the Results section to include an explanation of what each software is designed for and what is the input/output. Otherwise, the rationale and meaning of the analyses remain cloudy.

- 3) Related to the previous point: the methodology used is not clearly presented, and it is not clear where the isolates vs the WGS metagenomics reported in Suppl. Fig. 1 are used and analysed. Please clarify throughout the manuscript which data are at the basis of each analysis. The Methods section really needs an expansion to ensure reproducibility.
- 4) Again on reporting the methods used: Lines 251-254, how many genomes per clade did the authors analyse? There is a specification of how clades were defined as "commensal" or "invasive", but no definition for "intermediate" neither in the Methods nor in the Results (please add it to Fig. 5 caption as well).
- 5) Lines 301-303: did the authors test whether the plasmid was there? Since these are only 3 isolates, you could maybe check or remove the sentence about the virulence factors being encoded by a plasmid.
- 6) Lines 308-309: It is not clear what the authors mean here by the fact that unknowns cannot be trusted. I agree in principle, but did they consider unknowns not to be trusted also in other analyses or did they consider them as "good" alleles? did they check against reference lists for virulence factors?
- 7) confusing terminology: the UK and Norwegian genomic cohorts are sometimes called with the name of the country and some other times with the acronym. Please, define these cohorts in the Methods and at the beginning of the results as well, by naming them either with Country or acronym and stick with that defined name throughout the paper.
- 8) Numbers are not clear throughout the paper. Please report the cardinality of genomes used in each analysis to let the reader evaluate the relevance of the analysis.
- 9) Some references are wrong. For instance, reference 18 was mentioned at the beginning of the results, which is the right one. I guess 31 in Line 448 is wrong? I have some doubts about other genomic cohorts' references, please check throughout the manuscript.
- 10) Figures reported here are quite space-consuming and not necessarily the best way to convey the info. Some specific points:
 - A) Figure 1. for a better understanding, it would be great to have only half of each of the two plots (since they are mirrored, there's no need for the other half that only adds confusion) and even better to combine them to be able to compare line-wise vaginal and c-section delivered babies. By the way, this figure is quite informative and easy to understand.
 - B) Supplementary Fig. 2: This figure is particularly chaotic and is not easy to get the info the authors report. I don't see the logic behind the horizontal scale (I can see there are "random" timepoints where samples were taken, but for instance, days 8, 17 and 18 have no samples there? Maybe the scale should be changed or the rationale explained. Moreover, it is difficult to understand why there are different colours for the "after 21 days" and "after 4-12 months", which only adds confusion. If the authors would like to highlight the 4-12 months persistence, they could add a color or a symbol after the single ST name, or maybe color the shapes in the graph. If colour has to be used for lines, I would use it to make the persistence vs displacement more visible, or keep it only for the maternal source. Looking at the figure, it seems that no maternal strain is maintained in infancy. Is this the case? Not clear here, but there is space for improvement!

C) Suppl. Figure 1 is misleading and covers work performed in both the previous paper (Shao, Yan, et al., Nature 2019) and the present one. I would suggest simplifying the figure for readability and to use two different background colors or similar strategies to highlight what was done in the previous study (cohort enrollment, sample collection, cultivation, extraction and sequencing) from what was done in the present one (mSWEEP/mGEMS analyses, strain-level profiling, downstream analysis). In general, the study design is not well explained and presented in the manuscript, even though there is a long description of the cohort and study design of the previous published work. Additionally, figure quality and readability can be greatly improved:

- why has one of the arrows a different color? I guess it is an error?

- the dashed line under the "house" is different in size and colour

- why does "Reference data" connect with the "WGS metagenomics"? it should probably connect with the analyses?

D) Figure 4: Given the results, this could be a boxplot with "persistence" for vaginal and C-section for each ST. These figures are very space-consuming and not very useful because it is impossible to discriminate between two close colours on a 12-colours scale. The fancy-included plots could be supplementary.

E) Figure 5: it is not clear which are the 10 most frequent lineages in Norwegian bloodstream infections and which are those with the significant OR. Please check the figure and the caption.

F) Figure 5: with this method, the analysis should be performed by clade and not by ST, as both over and underestimation is possible. For instance, ST131 is intermediate according to Fig. 5A and clade C2 is instead invasive according to Fig. 5B. I would suggest either performing a more in-depth analysis also of the other STs or removing the ST-specific analysis. Otherwise, a more detailed discussion of this intra-ST heterogeneity is needed.

G) Figure 7: This color scale is ridiculous, how can one tell colours apart? :) See also comment to Figure 4.

H) Figure 8 is unreadable, the numbers on top are difficult to read even at 200%. Increase all font sizes, or consider using another graph to depict the same info

I also have some minor issues:

- Line 27: I don't think the analyses reported here are a "quantification of competition", as they are co-presence and co-exclusion analyses. I agree that the use of the word "competition" vs "co-exclusion" may help with readability, but no quantification has been performed in this study. Please rephrase.

- Supplementary Table 1 could probably just be an appendix or a section in the Methods

- Lines 181-182: what about the other 22 (1/4 of the total)?

- Lines 198-199: why not a lower abundance strain of maternal origin? even from other body sites

- Lines 376-380: why should this be in the gut? why not consider the skin or the oral microbiome? *K. pneumoniae* is usually present on the skin and upper respiratory tract, both of which could be sources of pioneering species in the gut.

- Lines 390-391: reference?

- remove references to figures in the Discussion

- Lines 412-419: please remove or blend this paragraph with the surrounding ones, as this has nothing to do with the analyses reported here. I guess the authors wanted to highlight that their method can be used also for other applications, but there's no discussion in this terms.

Dear Editor,

We would like to thank the reviewers for useful comments and suggestions that enabled us to revise the manuscript and thus significantly improve the study presentation. Below we provide a detailed response to each comment and how the manuscript has been revised accordingly.

Response to Reviewer #1

Major comments:

1. While the manuscript is entitled “strong pathogen competition in neonatal gut colonisation” the paper reads more as a “proof of concept” for the proposed methodology (combining metagenomics with strain-level genomic epidemiology). I would propose either to change the title to reflect the proposed methodology or expand and dedicate a paragraph on the neonatal microbiome in the introduction. In any case it should be clarified why the selected cohort (why the neonatal microbiome) was chosen.

Response: In retrospect it appears clear that the motivation to focus on this particular cohort was not appropriately communicated. We chose the particular neonatal cohort for multiple reasons. Firstly, because of its longitudinal aspect which enabled us to investigate dynamics of the competition. Secondly, due to its high sequencing depth and state-of-the-art short-read technology used, which enabled us to assemble the genomes of the colonising strains in sufficient detail for genomic epidemiological analysis. Thirdly, the DNA obtained from a sufficiently large number of faecal samples via a culture-independent approach provided an opportunity to screen for the pathogens in an unbiased and robust manner. Finally, the neonatal cohort was ideal for the study since it enabled us to probe the competition dynamics between the pathogen strains and species when they colonise a niche that is approximately empty, unlike the situation later in life. In addition, we screened a large body of microbiome to identify other published studies that would be suitable for our purposes, but only the study by Shao et al. was deemed appropriate and satisfying all these desiderata. In

the revision we have chosen to follow the latter suggestion to include a dedicated paragraph clarifying this in the Introduction section.

2. Directly related to the previous comment. Several conclusions are made in the study regarding the neonatal microbiome, including comparison between birth-mode. However, no discussion or comments are made regarding on the importance of these findings or its implication. Just one example, several Klebsiella species are found dominant in the vaginal delivery cohort, however what implications does this have on the development of the neonatal microbiome, or health later in life.

Response: As our study purpose was to unravel the gut colonisation competition dynamics for key human pathogens, we made no comments of the implications of the findings for the actual host health, either during infancy or later in life (of which we would not have any data available as the original study participants were anonymised). It is important to note that none of these pathogens are expected to cause disease in young children, unless they were immunocompromised or underwent surgery which would significantly increase the risk of the bacteria entering the bloodstream. We have added this comment to the Discussion. Otherwise, implications of our findings from the pathogen perspective are generally discussed in paragraphs 2-6 in the Discussion.

3. While the manuscript mentions the “population-wide quantification of strain-level colonisation dynamics for many common bacterial pathogens...”, the study seems to primarily focus on Enterococcus and Enterobacteriaceae, specifically colonisation of different Klebsiella species and E. coli. Was it decided beforehand to focus on these species, or was a comprehensive analysis done before hand and these were the only species of interest found within the neonatal microbiome? In any of the two scenarios please describe so within the manuscript.

Response: Indeed, our intended focus was on pathogenic Enterobacteriaceae and Enterococci, which have been widely considered in colonisation experiments in animal models and where availability of high-quality reference genomes is optimal. However, we did also screen for multiple other pathogen species but found only sporadic cases of

colonisation that would prevent drawing any robust conclusions regarding them. We have added information about the number of reliably identified colonisations for all species screened to the Results section in the form of Table 2 (previously Supplementary Table 1 with only the names and abbreviations of the target pathogens, but now with the numbers included).

4. The manuscript lacks any clear description of the used cohort. Although in different points in the manuscript different numbers of samples and timepoints are given this only provides more confusion since it is unclear how many samples/timepoints are present, nor how many samples in which timepoint. It may be a good idea to add a clear description at the start of the result section “Lineage-level analysis of neonatal gut microbiome data” and include the nr. of samples of each cohort/group at each individual timepoint. Maybe include a summary figure.

Response: We agree that this is very helpful information to the reader and have added it in the Results.

5. An antagonistic relationship between E.coli and different Klebsiella species has been reported in the results (L127-131). Although significant ($p < 0.05$), the correlation between said pathogens ranged to max -0.2. I’m not very well versed in cut-offs for correlations, but would you still consider this antagonistic?

Response: Correlation values are generally diluted in this kind of a setting, because very low levels of abundance can in practice lead to part of the data indicating non-existing relationships. Hence, high absolute values of a correlation coefficient cannot be expected, despite that the analyses presented later in the article indicate fairly strong antagonistic relationships between particular species. In addition, experimental work cited by us demonstrate such relationships for some of the species pairs considered here.

6. While the methods section is overall extensive, please provide additional, and if possible, code, of the specific settings used in the analyses. For instance, the script mentioned in L460. In L478-482, different options were used and the result with both high-quality metrics

and was restively consistent with the species; MLST designations. Please, provide additional information and note which option was finally chosen. Additionally, different tools such as mSWEEP and mGEMS was used, for the reader who may not be familiar with these tools it may be interesting to include a description of these tools.

Response: We have added the script mentioned in L460 and the slurm batch job scripts used to run the analyses to the GitHub repository containing the code for creating the visualisations (<https://github.com/tmaklin/baby-microbiome-paper-plots>). The batch job scripts contain the code/specific settings used for each program but we would like to clarify that unfortunately they cannot be run as-is to fully reproduce the analyses since the same script was used with different inputs to perform the analyses in several parts due to the considerable resources required to run all parts of the analyses at once. We have added the scripts with the purpose of documenting the different options used for each program.

We have also revised the methods section to include a short description of the different tools used, and included information about which option was chosen on lines (comment regarding L478-482) in Table 3 under the Reference sequence grouping subsection of the revised manuscript.

7. The used approach is described as a “combination of latest high-resolution metagenomics and strain-level genomic epidemiology methods leveraging large genomic reference libraries”. To fully appreciate this methodology could you please include a description of said method would include and how this would differ from any other reference sequence based approach?

Response: The primary difference between our approach and other methods is that we characterise the abundant pathogen species at the level of internationally circulating clones and lineages, and assemble their genomes from metagenomic sequencing data to allow for further genomic epidemiological analysis, such as use of phylogenetic trees. We have added this clarification into the revision (early in the Discussion).

Minor comments:

8. L92. mentions “de-mixing metagenomes”, please provide a description on what could be understood of this.

Response: We have removed the term “de-mixing” here and elsewhere to avoid confusion.

9. L99. “unprecedented depth”, please mention what depth this would be.

Response: Information added.

10. For figure 2, I would advise to create two different panels representing each in turn the absolute and relative differences. As it is, the figure is a bit confusing.

Response: We have split the figure into two panels as suggested.

11. L166. 6 times out of how many? Please provide some additional information.

Response: Information added.

12. L169-L189. While interesting, this section is a bit confusing to read. For a large part this is since the cohort has not been properly introduced before (comment 4). For readability, it may further be good to include a concise notation of the different timepoints and the relevant results at each time point. Additionally, for clarity it may be good to use percentages instead of numbers. i.e. 28% instead of 88/314.

Response: We have now improved readability by introducing the cohort as suggested in the earlier major comment.

13. L190-L194. Again, to improve clarity, it would be better to first report on the mothers (43 out of how many?) and then note that 16 children of these mothers were found to share the same lineage.

Response: We have added the total number of mothers in the study cohort (175) here and introduced the cohort earlier in the manuscript as suggested.

14. L202-203. Please provide numbers.

Response: Numbers added.

15. L206-L207. Which timepoints would this be?

Response: These lines refer to the chronologically last point in each individual's time series from the first 21 days, where *E. coli* was detected. We have clarified the phrasing to make this clearer.

16. L209. You may have given this earlier, but please provide again the number of samples which were observed to contain multiple E.coli lineages.

Response: Numbers added.

17. L200-L220. Although supplementary figures are provided for this section, the results may be cleared and better to follow if a (summary) figure was included within the main manuscript.

Response: We have added a summary figure in the main manuscript and improved the supplementary figure. Thank you for the suggestion.

18. L227-L228. Please provide a paragraph explaining the event matrix.

Response: We have added the following sentence explaining the event matrix to the paragraph: "The event matrix contains the numbers of times a transition from carriage of a specific lineage (rows) to carriage of another lineage or persistence of the same lineage (columns) was observed."

19. L237. Please provide a short description of the odds ratio and how this was calculated.

Response: This information was already provided in the Methods in a separate section titled 'Odds ratios for invasiveness'. Since the odds ratio is a fundamental epidemiological concept we believe it is sufficient to refer to the Methods (now added in text) and provide a citation for a reader who is not familiar with it.

20. L236-L257. Please provide additional description of why said cohorts were chosen. For L247. Would that be the prevalence in carriage within the neonatal dataset?

Response: These two cohorts are the largest, and most importantly systematic and representative (unbiased) genomic studies of E. coli BSI from numerous hospitals in the two countries, thus providing the necessary resolution for comparing frequencies in colonisation and disease. We have now added this motivation to the text.

21. Figure 8 is quite difficult to decipher. Please consider replacing the plot with a table, or else increase the size.

Response: We have increased the font size in Figure 8 (Figure 9 in the revised manuscript) to a more readable level.

22. To identify AMR within the klebsiella, have you used any other tools than Kleborate? I'm not familiar with Kleborate, but I know for instance that rgi from the card-database has demonstrated better performances than AMRFinderPlus from NCBI.

Response: Kleborate is the state-of-the-art tool for genomic profiling of Klebsiella isolates, including AMR and virulence determinants. Its reference database has been carefully curated to optimise performance for this genus so we do not expect that other tools would provide additional insight. We have added a clarification about our rationale for using Kleborate in the Detection of AMR and virulence genes in *Klebsiella* strains section.

23. L298-L312. Speaks about “isolates”, while in L289 “assemblies” are mentioned. In this section (L287 and down), where specific isolates used or reconstructed genomes from the metagenomic samples?

Response: We apologise for the sloppy and misleading phrasing. All “isolates” refer to those identified from the metagenomic data by our approach, and are thus not isolates in the technical sense. We did not use specific isolate data from Shao et al. in the characterisation of the samples, only strain assemblies from metagenomes. The text has now been corrected here and elsewhere.

24. L330. Please define “novel methodology”

Response: We have rephrased to remove ambiguity.

25. L342. “common pathogenic gut bacteria”, in this case would be *Klebsiella* species in specific?

Response: We have rephrased to remove ambiguity.

26. L391-396. Please provide additional information on the cohorts. How many samples, etc. Please also provide a description of what you would consider “relative invasiveness” and how you would determine this.

Response: The additional information about the cohorts has been added to the Methods as suggested in an earlier comment. By “relative invasiveness” we referred to the ORs, this is now rephrased to remove ambiguity.

Response to Reviewer #2

Summary (Reviewer #2)

Mäklin and colleagues present an investigation of the prevalence, colonization potential, and competition dynamics of pathogenic gut species in a previously published cohort of gut metagenomes and isolates obtained from infants born by vaginal or C-section delivery. They highlight some co-presence and co-exclusion patterns of noteworthy pathogens, expose the dynamics of *E. coli* strains colonization in the newborns' gut (including the rare strain substitution), and evaluate the invasiveness of selected STs in light of larger cohorts. Additionally, they rule out that *E. faecalis* strains colonizing the newborns' gut are the long-term persistent hospital-adapted ones and survey the antimicrobial and virulence repertoire of their *Klebsiella* strains.

Despite the great amount of data to analyse from the previous publication, this work fails to address key questions in the field like exposing the maternal route for the colonization events (there are mothers' samples that are also presented or even highlighted here and there but never properly investigated and discussed) and presents a very shallow biological understanding and discussion of the results of the performed computational analyses. Moreover, the results presented here are not discussed in light of the enormous amount of literature in the field, and the quality/curation of

the figures is low. Methods are not clear enough to ensure reproducibility, and there are many convoluted parts. Nevertheless, the content of this work might be of interest to a more specialized audience.

Major issues:

1) The authors state that they are surveying to what degree E. coli is inherited from the mother (Line 173) but they only report lineage sharing data. Such analysis should be done by comparing strains carried by the mother with those carried by the newborn, and to me, it is not clear why the authors did not attempt this (especially because they have isolates as well). On this line, the authors use the word "strain" multiple times in the text to refer to lineages, I guess? Otherwise, there is no explanation of how they performed the strain-level analyses in the Methods. I understand that this might only be a matter of the use of the language, but please clarify.

Response: We apologise for the lack of clarity. Indeed, “strain” refers to a specific lineage identified in genomic epidemiological studies and typically corresponds either to an MLST sequence type (such as E. coli ST131) or a subtype (such as ST131-A). We have now revised the text in the beginning of the Results and notify the reader about this convention to avoid ambiguity. Regarding the samples from mothers, we have now included a more systematic analysis of the mother vs infant pairs of strains and a new figure illustrating the results. What comes to the isolate data from the original study, we decided not to include them in the analysis of the within-sample variation to avoid biasing our conclusions. This is due to the fact that isolates were cultured in a highly uneven manner, such that a majority of them represented Enterococcus (n = 451), followed by Klebsiella spp. (n = 235), while very small fraction was available for the Escherichia spp (n = 41). As E. coli was found to be the most dominant species in many samples by the metagenomic analysis, isolate data would not have covered samples in a systematic fashion. Despite that more isolates were available for Enterococci, it was well motivated to treat all species in the same manner in our analysis. We have now also clarified these aspects in the revision. Finally, we added a clarification in the Discussion why a review of our results in the

context of more general microbiome literature was out of the scope of our study.

2) In general, this manuscript is of interest mainly to clinical microbiologists, who might not be familiar with the proposed methodologies developed by the authors in previous publications. I would suggest expanding the Methods but also the Results section to include an explanation of what each software is designed for and what is the input/output. Otherwise, the rationale and meaning of the analyses remain cloudy.

Response: We have added this information in the revision. In addition we would like to highlight that the results would likely be of more general interest, for example to microbiologists working on experimental evolution and on competition between bacteria from the ecological perspective.

3) Related to the previous point: the methodology used is not clearly presented, and it is not clear where the isolates vs the WGS metagenomics reported in Suppl. Fig. 1 are used and analysed. Please clarify throughout the manuscript which data are at the basis of each analysis. The Methods section really needs an expansion to ensure reproducibility.

Response: We apologise for this ambiguity and have now added all requested information. As a note, isolate WGS data from the neonatal microbiome study were not used elsewhere than as a part of the reference collection in the mSweep/mGems method pipeline. We have clarified the use of these two datasets in the “Sequencing data” subsection under the Methods section and in Supplementary Figure 1.

4) Again on reporting the methods used: Lines 251-254, how many genomes per clade did the authors analyse? There is a specification of how clades were defined as "commensal" or "invasive", but no definition for "intermediate" neither in the Methods nor in the Results (please add it to Fig. 5 caption as well).

Response: We have added the number of genomes per clade and clarified the definition of the “commensal” and “invasive” clades, and added the missing definition of the “intermediate” clades, which are the clades where the

confidence interval in Figure 5 (Figure 6 in the revised manuscript) contains the value 1.

5) Lines 301-303: did the authors test whether the plasmid was there? Since these are only 3 isolates, you could maybe check or remove the sentence about the virulence factors being encoded by a plasmid.

Response: We checked for the presence of this plasmid by mapping the assemblies to the reference plasmid sequence, and in each of the 3 isolates >90% of bases mapped, indicating that the plasmid was present. We have updated the text accordingly.

6) Lines 308-309: It is not clear what the authors mean here by the fact that unknowns cannot be trusted. I agree in principle, but did they consider unknowns not to be trusted also in other analyses or did they consider them as "good" alleles? did they check against reference lists for virulence factors?

Response: We understand that this was unclear and have added some clarification to show that the genes were genuinely detected (> 90% nucleotide identity and > 80% coverage), but that the alleles were different to those found in clinical strains in *K. pneumoniae*. We also cite two studies that show ybt to be relatively common in the more environmental *Klebsiella* species, so we are confident that it is a real observation (and therefore worth noting), but that more research is needed to understand the evolutionary and clinical significance of these loci.

7) confusing terminology: the UK and Norwegian genomic cohorts are sometimes called with the name of the country and some other times with the acronym. Please, define these cohorts in the Methods and at the beginning of the results as well, by naming them either with Country or acronym and stick with that defined name throughout the paper.

Response: We agree and do now recognise the risk of confusion here. The definitions have been added as suggested and to have maximal clarity the country information has been combined with the surveillance system acronym (NORM in Norway and BSAC in the UK).

8) Numbers are not clear throughout the paper. Please report the cardinality of genomes used in each analysis to let the reader evaluate the relevance of the analysis.

Response: Excellent suggestion, we have now added this information.

9) Some references are wrong. For instance, reference 18 was mentioned at the beginning of the results, which is the right one. I guess 31 in Line 448 is wrong? I have some doubts about other genomic cohorts' references, please check throughout the manuscript.

Response: Reference #31 is not wrong, but actually contains also the genomes of reference #18. In our study we have further improved the resolution of the E. coli reference by adding the later published 3,254 genomes from Gladstone et al. (reference #17) to this extensive curated reference collection (please note that the numbers have changed in the revision due to added references).

10) Figures reported here are quite space-consuming and not necessarily the best way to convey the info. Some specific points:

A) Figure 1. for a better understanding, it would be great to have only half of each of the two plots (since they are mirrored, there's no need for the other half that only adds confusion) and s n better to combine them to be able to compare line-wise vaginal and cs section delivered babies. By the way, this figure is quite informative and easy to understand.

Response: We have combined the two panels into a single-panel figure, where the lower triangle of the matrix shows the correlations for the vaginal delivery cohort and the upper triangle the caesarean delivery cohort. Thank you for the great suggestion!

B) Supplementary Fig. 2: This figure is particularly chaotic and is not easy to get the info the authors report. I don't see the logic behind the horizontal scale (I can see there are "random" timepoints where samples were taken, but for instance, days 8, 17 and 18 have no samples there? Maybe the scale should be changed or the rationale explained. Moreover, it is difficult to understand why there are different colours for the "after 21

days" and "after 4-12 months", which only adds confusion. If the authors would like to highlight the 4-12 months persistence, they could add a color or a symbol after the single ST name, or maybe color the shapes in the graph. If colour has to be used for lines, I would use it to make the persistence vs displacement more visible, or keep it only for the maternal source. Looking at the figure, it seems that no maternal strain is maintained in infancy. Is this the case? Not clear here, but there is space for improvement!

Response: We agree that this figure is quite difficult to read, however, it contains information that is otherwise challenging to summarize visually. We have revised the figure to improve interpretability. We have also added a summary figure in the relevant section of the manuscript (Neonatal gut colonisation of *E. coli* adheres to the first-come, first-served principle) which hopefully further aids in interpreting the results.

C) Suppl. Figure 1 is misleading and covers work performed in both the previous paper (Shao, Yan, et al., Nature 2019) and the present one. I would suggest simplifying the figure for readability and to use two different background colors or similar strategies to highlight what was done in the previous study (cohort enrollment, sample collection, cultivation, extraction and sequencing) from what was done in the present one (mSWEEP/mGEMS analyses, strain-level profiling, downstream analysis). In general, the study design is not well explained and presented in the manuscript, even though there is a long description of the cohort and study design of the previous published work. Additionally, figure quality and readability can be greatly improved:

- why has one of the arrows a different color? I guess it is an error?*
- the dashed line under the "house" is different in size and colour*
- why does "Reference data" connect with the "WGS metagenomics"? it should probably connect with the analyses?*

Response: We have revised Supplementary Figure 1 according to the suggestions provided and apologise for the errors in the previous version.

D) Figure 4: Given the results, this could be a boxplot with "persistence" for vaginal and C-section for each ST. These figures are very space-consuming and not very useful because it is impossible to discriminate between two close colours on a 12-colours scale. The infancy-included plots could be supplementary.

Response: We have clarified the results in the figure by adding the numbers of transitions observed on top of each coloured box in the plot and by moving the infancy-included plots into the supplementary as suggested. Since the values represented by our plot are discrete counts of transitions from carriage/persistence of some lineage(s), we do not think a boxplot (typically used to display categorised continuous observations or a large number of categorised discrete observations) is necessarily a good way to present our results. We hope that with the other suggested changes the figure is satisfactory.

E) Figure 5: it is not clear which are the 10 most frequent lineages in Norwegian bloodstream infections and which are those with the significant OR. Please check the figure and the caption.

Response: We added the list of the 10 most frequent lineages in the NORM collection to the figure caption.

F) Figure 5: with this method, the analysis should be performed by clade and not by ST, as both over and underestimation is possible. For instance, ST131 is intermediate according to Fig. 5A and clade C2 is instead invasive according to Fig. 5B. I would suggest either performing a more in-depth analysis also of the other STs or removing the ST-specific analysis. Otherwise, a more detailed discussion of this intra-ST eterogeneity is needed.

Response: This is an in principle valid comment, however, please note that Fig 5A uses only the NORM disease cohort, while the 5B shows ORs for both BSAC (UK) and NORM (Norway) disease cohorts, where the conclusions from the latter are not different from those in 5A regarding ST131. We have now revised the text and the figure to avoid ambiguity of interpretation. The other major clinically relevant E. coli STs do not have similar internationally

established clonally expanding clades with recognized nomenclature as ST131 does. Therefore, there is no firm epidemiological basis for performing such an analysis outside ST131. Earlier studies have not demonstrated the existence of deep branching clades in the other major STs, and hence it is meaningful to keep the OR analysis at the ST level for them.

*G) Figure 7: This color scale is ridiculous, how can one tell colours apart?
:) See also comment to Figure 4.*

Response: We have made the same changes to Figure 7 that we made to Figure 4 as suggested by the comment (Figures 8 and 5 in the revised manuscript).

H) Figure 8 is unreadable, the numbers on top are difficult to read even at 200%. Increase all font sizes, or consider using another graph to depict the same info

Response: We have increased the font sizes and apologise for the ridiculously small fonts in the previous version.

Minor issues:

- Line 27: I don't think the analyses reported here are a "quantification of competition", as they are co-presence and co-exclusion analyses. I agree that the use of the word "competition" vs "co-exclusion" may help with readability, but no quantification has been performed in this study. Please rephrase.

Response: We agree and have rephrased accordingly.

- Supplementary Table 1 could probably just be an appendix or a section in the Methods

Response: We have moved Supplementary Table 1 to the Results section and included additional information about the number of times the species were identified in the WGS metagenomics data.

- Lines 181-182: what about the other 22 (1/4 of the total)?

Response: We have added information about the other 22 cases to the manuscript. In the remaining 22 individuals, in 3 cases the lineage identified at 4 days had been replaced by another lineage and in 19 cases no *E. coli* was present in detectable amounts.

- Lines 198-199: why not a lower abundance strain of maternal origin? even from other body sites

Response: We have now revised the text to take this into account and make a more cautious interpretation of the results.

- Lines 376-380: why should this be in the gut? why not consider the skin or the oral microbiome? K. pneumoniae is usually present on the skin and upper respiratory tract, both of which could be sources of pioneering species in the gut.

Response: We have now revised the text to take this into account and make a more cautious interpretation of the results.

- Lines 390-391: reference?

Response: We have reworded this to remove ambiguity and added published examples illustrating the role of non-maternal sources.

- remove references to figures in the Discussion

Response: Done.

- Lines 412-419: please remove or blend this paragraph with the surrounding ones, as this has nothing to do with the analyses reported here. I guess the authors wanted to highlight that their method can be used also for other applications, but there's no discussion in this terms.

Response: Thanks for pointing out that this text was disconnected from the current analyses. We have now edited the text to bring in the relevant connection.

REVIEWERS' COMMENTS

Reviewer #2 (Remarks to the Author):

The Authors replied to all comments and issues raised by me and the other Reviewer in a very satisfying manner.

Figures, nomenclature, and numbers, which were the most confusing parts of the paper, have been fixed and the flow and rationale of the paper are now much easier to follow.

I have no further issues.